# What Makes a Good Feedforward Computational Graph?

**Alex Vitvitskyi** [1]  **João G.M. Araújo** [1]  **Marc Lackenby** [2]  **Petar Veličković** [1]

## Abstract

As implied by the plethora of literature on graph rewiring, the choice of *computational graph* employed by a neural network can make a significant impact on its downstream performance. Certain effects related to the computational graph, such as under-reaching and over-squashing, may even render the model incapable of learning certain functions. Most of these effects have only been thoroughly studied in the domain of *undirected* graphs; however, recent years have seen a significant rise in interest in *feedforward* computational graphs: directed graphs without any back edges. In this paper, we study the desirable properties of a feedforward computational graph, discovering two important complementary measures: *fidelity* and *mixing time*, and evaluating a few popular choices of graphs through the lens of these measures. Our study is backed by both theoretical analyses of the metrics' asymptotic behaviour for various graphs, as well as correlating these metrics to the performance of trained neural network models using the corresponding graphs.

## 1. Introduction

Modern deep learning workloads frequently necessitate processing of *sequential* inputs, such as words in a sentence (Sutskever et al., 2014), samples of an audio recording (Van Den Oord et al., 2016), partially ordered nodes in a graph (Thost & Chen, 2021), execution steps of an algorithm (Veličković et al., 2022), or snapshots of edits made to temporal graphs (Rossi et al., 2020). In addition, many important self-supervised learning tasks require efficiently predicting future evolution of such inputs, with examples ranging from temporal link prediction (Huang et al., 2024) to next token prediction (Radford et al., 2018).

In regimes like this, it is important to be able to train such

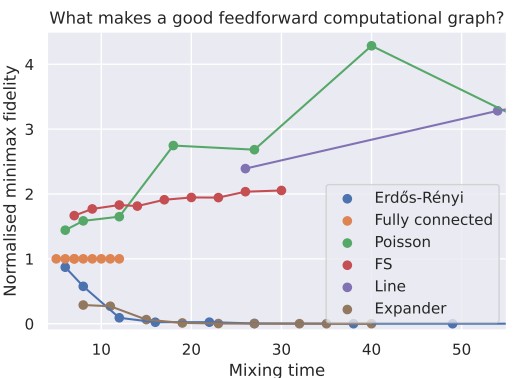

What makes a good feedforward computational graph?

*Figure 1.* In this paper, we propose two measures which can be used to assess suitability of *feedforward* computational graphs for neural networks – **mixing time** (lower is better; Sec. 4) and (normalised) **minimax fidelity** (higher is better; Sec. 5). Here we plot these metrics as attained by various graph generators—at each point in a sequence, the number of nodes doubles, starting from 16. Using FunSearch (Romera-Paredes et al., 2023), we discover the **FS graphs**, which have favourable $O(\text{polylog } n)$ mixing time, while retaining higher fidelity compared to a fully connected graph (Sec. 6). Several of these graphs may be viewed in Appendix D.

models *scalably* and *without leaking* ground-truth data about the future input parts that need to be predicted. As such, many modern architectures resort to *feedforward* computational graphs, wherein information may only flow from older samples towards newer ones—never in reverse! Applying such a graph – also known as a "causal mask" (Srambical, 2024) – allows for scalable training over entire chunks of the input at the same time.

This naturally invites the question: *what makes a good feedforward computational graph?* Alternately worded, which considerations need to be taken into account when deciding which feedforward graph to use (see Figure 1)?

Here we propose two suitable and complementary ways to measure suitability of feedforward graphs: **mixing time**—the speed at which input data converges towards a stationary distribution—and **minimax fidelity**—the sharpness of the data is as it propagates through the graph. We supplement our measures with thorough theoretical derivations across several graph generators, and correlate them to empirical performance, paving the way to future studies in the area.

[1]Google DeepMind [2]University of Oxford. Correspondence to: Alex Vitvitskyi <avlife@google.com>.

*Proceedings of the $42^{nd}$ International Conference on Machine Learning*, Vancouver, Canada. PMLR 267, 2025. Copyright 2025 by the author(s).

Note that by "feedforward" we imply a statement about our *input data*—not the *neural network* processing it! That is, the assumption that our input nodes have a sequential ordering to them which must be respected when processing them—see Section 3. We therefore do not design our framework with more general feedforward neural networks (such as MLPs) in mind—however, we believe some of the ideas explored here could be used to study propagation in such unrestricted networks as well.

## 2. Motivation

We were inspired to seek an answer to this question, given that there is already a rich, extensive body of work on studying undirected computational graphs (commonly referred to as *graph rewiring*). Therein, important issues such as over-smoothing (Li et al., 2018; Oono & Suzuki, 2019; Keriven, 2022), oversquashing (Alon & Yahav, 2021; Di Giovanni et al., 2024) and under-reaching (Barceló et al., 2020) have all been identified, and related to the input graph topology. In response, a substantial amount of methods have been proposed to modify the provided input graph for improved diffusion (Gasteiger et al., 2019), curvature (Topping et al., 2022; Fesser & Weber, 2024), effective resistance (Arnaiz-Rodríguez et al., 2022), or reducing smoothing (Azabou et al., 2023) and commute time (Sterner et al., 2024). While efforts were made to generalise these techniques to directed graphs (Maskey et al., 2023), as well as for sparsifying attention-based architectures (Zaheer et al., 2020; Liao & Póczos, 2024), in all previous cases they focussed on graphs where backwards edges are explicitly allowed.

Eventually, the focus of study expanded beyond "how to modify an input graph to be better?" towards "what makes a good input graph?", propelling the discovery of *task-agnostic* graphs that are guaranteed to have good topological properties. This question is well-studied in mathematics, where it had led to the advent of *expander graphs* (Kowalski, 2019); graphs with highly sparse topologies but incredibly good information propagation. Once expanders have been discovered in the context of graph machine learning, they have seen equal application among graph neural networks (Deac et al., 2022; Christie & He, 2023; Wilson et al., 2024) and graph Transformers (Shirzad et al., 2023; 2024).

Unfortunately, to the best of our knowledge, the state-of-the-art is not as rich in the domain of feedforward graphs. This includes the domain of mathematics, wherein many important concepts have not been generalised to the directed case[1] as they may rely on spectral properties of the graph structure (Chung, 1997), and many spectral properties are ill-defined

on a feedforward graph. In terms of practical usage, most of the heavy lifting is done either by *fully connected* feedforward graphs or *locally-connected* sliding-window graphs. And while recent work has identified limitations of high-indegree feedforward graphs through over-squashing (Barbero et al., 2024) and dispersion (Veličković et al., 2024), most such works do not actively offer a different computational graph structure, nor do they offer any principles that can be used to derive one. We seek to fill this gap.

**What's in a good metric?** Both of the above papers make it clear—at least if size generalisation is a desirable property—that it is beneficial to *limit the in-degree of each node in the feedforward graph.*

A good metric should be able to help us, in the very least, **compare against different graph distributions with *same asymptotic in-degree budget*.**

It might also be helpful if the metric would be able to hint to practitioners at what point the in-degree budgets become **problematic**—specifically, this means they shouldn't be optimised for the fully connected graph. This is not strictly necessary, as the papers above already provide ample proof.

## 3. Theoretical Foundations

In this work, we study **feedforward graphs**, $\mathcal{G} = (\mathcal{V}, \mathcal{E})$, i.e., graphs where nodes are given by an ordered set of $n$ integers ($\mathcal{V} = \mathbb{Z}_n$) and edges are constrained to only go *forwards*; that is, $(a, b) \in \mathcal{E} \implies a \leq b$. We will denote by $\tau \in \mathcal{V}$ the *sink vertex* of that graph ($\tau = n - 1$), a node which is used to drive decision-making, and which is not allowed to have any outward edges (except to itself).

Much like a degree in an undirected graph, a node $i$ in a feedforward graph has an *indegree* $\delta_{i\leftarrow}$ and an *outdegree* $\delta_{i\rightarrow}$, counting the number of incoming and outgoing edges to/from node $i$.

As an illustrative example, two types of feedforward graph are commonly used for machine learning tasks today:

- The **fully connected** feedforward graph, which draws an edge between every allowed pair of nodes; $\mathcal{E} = \{(a, b) \mid a \leq b\}$. Indegrees and outdegrees in this graph are $\delta_{i\leftarrow} = i + 1, \delta_{i\rightarrow} = (n - i)$.

- The **locally connected** feedforward graph, which draws an edge between all allowed pairs of nodes up to a distance $\kappa \geq 0$ apart: $\mathcal{E} = \{(a, b) \mid b - \kappa \leq a \leq b\}$. A special case of $\kappa = 1$ is known as a **line graph**. Indegrees and outdegrees are $\delta_{i\leftarrow} = \delta_{i\rightarrow} = \kappa + 1$ provided $\kappa \leq i \leq n - \kappa - 1$.

We now outline three desirable properties which we will typically assume during our exploration:

---

[1] We are aware of one prior work generalising undirected expanders to the concept of a feedforward *extender* (Csóka & Grabowski, 2022)—we leave making any connections of extender graphs to the propagation properties studied here for future work.

**Self-Edges.**  We will always assume all self-edges are present in the graph – that is, $(i, i) \in \mathcal{E}$ for all $i \in \mathcal{V}$, unless otherwise stated. This is generally an important design decision for improving data retention.

**Unique Sinks.**  As we will be tracking how easily information from each node can reach the designated sink node, $\tau$, it is highly desirable that our graph generator does not create additional sink nodes of any kind (beyond the final node, of course). Any additional sink nodes imply an impossibility to reach the final sink from such nodes, and they are hence effectively excluded from decision making.

**Self-Similarity.**  While most of our theory will concern the flow of information specifically into $\tau$, it is important to note that in real workloads, *any* node might be used as a sink node (over an appropriately sampled sub-input). As such, we focus our attention on *self-similar* feedforward graphs, wherein we can expect similar flow properties to all nodes (including ones in the middle).

# 4. Mixing Time: Tracking Path Complexity

We are now ready to work our way towards defining the first of our two measures, the *averaged mixing time* of the feedforward graph. This measure tracks how quickly information travels towards the sink, by carefully analysing the expected path length from each node to the sink. Clearly, the aim is to keep this value within reasonable upper bounds.

## 4.1. Basic Notions

For a feedforward graph of $n$ nodes, $\mathcal{G} = (\mathbb{Z}_n, \mathcal{E})$, let $\mathbf{A} \in \mathbb{R}^{n \times n}$ be its adjacency matrix, defined in the usual way:

$$a_{ij} = \begin{cases} 1 & (j, i) \in \mathcal{E} \\ 0 & (j, i) \notin \mathcal{E} \end{cases} \tag{1}$$

keeping in mind that $\mathbf{A}$ must be lower-triangular due to the feedforward constraint.

Its *walk matrix* $\mathbf{W}$ is given by

$$w_{ij} = \begin{cases} 1/\delta_{j\rightarrow} & (j, i) \in \mathcal{E} \\ 0 & (j, i) \notin \mathcal{E}. \end{cases}$$

So $\mathbf{W}$ is the transition matrix for the *(lazy) random walk* where there is an equal probability of leaving vertex $j$ along any of its outgoing edges. This will be well defined as long as $\delta_{j\rightarrow} > 0$ for all $j \in \mathbb{Z}_n$; we are guaranteeing this condition by requiring all self-edges within the graph.

Note that for graphs where, for all $j \in \mathbb{Z}_n$, $\delta_{j\rightarrow} = \kappa$ for some constant, $\kappa$, the walk matrix is related to the usual adjacency matrix $\mathbf{A}$. Specifically

$$\mathbf{W} = \frac{1}{\kappa} \mathbf{A}.$$

Hence, in this case, the spectrum of the walk matrix is obtained from the spectrum of $\mathbf{A}$ by scalar multiplication.

## 4.2. Stationary Distributions

A probability distribution $\boldsymbol{\pi} \in \mathbb{R}^n$ on the vertices is *stationary* if $\mathbf{W}\boldsymbol{\pi} = \boldsymbol{\pi}$. In other words, $\boldsymbol{\pi}$ is unchanged by one step of the random walk.

It is a well-known result that a strongly connected graph has a unique stationary distribution. Here, a graph is *strongly connected* if, for any ordered pair of vertices, there is an oriented path joining them. However, the graphs we will consider will be feedforward and hence not strongly connected. In that case, we have the following useful result.

**Lemma 4.1.** *Let $\mathcal{G}$ be a feedforward graph with a unique sink vertex $\tau$. Then there is a unique stationary distribution for $\mathbf{W}$, namely $\mathbf{1}_\tau$, the probability distribution taking the value $1$ at $\tau$ and $0$ elsewhere.*

*Proof.* Certainly $\mathbf{1}_\tau$ is stationary. To see that it is unique, consider a distribution $\boldsymbol{\pi}$ other than $\mathbf{1}_\tau$. Let $i < n - 1$ be its smallest vertex with $\pi_i \neq 0$. When applying the random walk matrix $\mathbf{W}$, we can see that the probability of being at $i$ after one step is $\pi_i/\delta_{i\rightarrow}$. Since $\delta_{i\rightarrow} > 1$, we deduce that $\boldsymbol{\pi}$ is not stationary. $\qquad\square$

## 4.3. Mixing Times

There are several definitions of mixing time in the literature. Here is one. The *mixing time* is the smallest value of $t$ such that for any starting distribution $\mathbf{x}$,

$$\|\mathbf{W}^t \mathbf{x} - \boldsymbol{\pi}\|_1 < 1/4.$$

Here, the $L^1$ norm is used on probability distributions, i.e., for two probability distributions $\mathbf{x}$ and $\mathbf{y}$,

$$\|\mathbf{x} - \mathbf{y}\|_1 = \sum_i |x_i - y_i|.$$

There is nothing special in the use of $1/4$ here. Any fixed $0 < \epsilon < 1/2$ would work. In our set-up, it is reasonable not consider a minimum over starting distributions $\mathbf{x}$ but an 'averaged' version:

$$\frac{1}{n} \sum_i \|\mathbf{W}^t \mathbf{e}_i - \boldsymbol{\pi}\|_1,$$

where $\mathbf{e}_i$ is the probability distribution concentrated on vertex $i$. If we let $\boldsymbol{\Pi}$ be the square matrix that has $\boldsymbol{\pi}$ in each column, then this is

$$\frac{1}{n} \|\mathbf{W}^t - \boldsymbol{\Pi}\|_1,$$

where now we are taking the $L^1$ norm on matrices:

$$\|\mathbf{B}\|_1 = \sum_{ij} |b_{ij}|.$$

We call this quantity the *averaged mixing time* (Espuny Díaz et al., 2024). We will now demonstrate its behaviour is as expected on two commonly used graph distributions.

### 4.3.1. LINE GRAPHS

Let $\mathcal{G}$ be the line graph from vertex 0 to vertex $n - 1$, as previously defined. Suppose that we start a random walk at vertex $n - i - 1$, i.e., the initial probability distribution is $\mathbf{e}_{n-i-1}$. We want to consider the probabilty of not reaching the terminal vertex after $t$ steps. We can think of flipping a coin $t$ times and we step right only if we get a head. We reach the terminal vertex if and only if we get at least $i$ heads. So the probability of not reaching it in $t$ steps is

$$\sum_{j<i} \binom{t}{j} 2^{-t}.$$

The average of this value, over all $i$ between 0 and $n - 1$, is

$$\frac{1}{n} \sum_{i=0}^{n-1} \sum_{j<i} \binom{t}{j} 2^{-t} = \frac{1}{n} \sum_{i=0}^{n-1} (n-i-1) \binom{t}{i} 2^{-t}.$$

When $n \geq t$,

$$\sum_{i=0}^{n-1} (n-1) \binom{t}{i} 2^{-t} = (n-1)$$

$$\sum_{i=0}^{n-1} i \binom{t}{i} 2^{-t} = \sum_{i=0}^{t} i \binom{t}{i} 2^{-t} = t/2$$

So, the average value of the probability of not reaching the terminal vertex is

$$\frac{1}{n} \left( n - 1 - \frac{t}{2} \right) = 1 - \frac{t+2}{2n}.$$

This is less than $1/4$ when $t > 3n/2$. Hence, the averaged mixing time is $3n/2$, which is $O(n)$.

### 4.3.2. FULLY CONNECTED GRAPHS

Now consider the case when $\mathcal{G}$ is the fully connected feedforward graph. For convenience, we will label the graph with vertices $\{0, \ldots, n\}$ where every pair of vertices $i$ and $j$ with $i < j$ are joined by an oriented edge from $j$ to $i$. (Hence, in this scenario, 0 is the terminal vertex.) Then the average mixing time is of the order $O(\log n)$.

To prove this, say that we start at position $i_0$ and then move to position $i_1$, ending at $i_t$ after $t$ steps. Then the expected value of the vertex after $t$ steps is

$$\frac{1}{n+1} \sum_{i_0=0}^{n} \frac{1}{i_0+1} \sum_{i_1=0}^{i_1} \cdots \sum_{i_{t-1}=0}^{i_{t-2}} \frac{1}{i_{t-1}+1} \sum_{i_t=0}^{i_{t-1}} i_t$$

$$= \frac{1}{n+1} \sum_{i_0=0}^{n} \frac{1}{i_0+1} \sum_{i_1=0}^{i_1} \cdots \sum_{i_{t-1}=0}^{i_{t-2}} \frac{1}{i_{t-1}+1} i_{t-1}(i_{t-1}+1)/2$$

and inductively, we can show that this is

$$\frac{n}{2^t}.$$

When $t > \log_2 4n^2$, this expectation is at most $1/(4n)$. This implies that the probability of being on the terminal vertex 0 is at least $3/4$. Hence, the mixing time is at most $\log_2 4n^2 = 2 + 2 \log n$.

## 4.4. Discussion About Mixing Time as a *Good* Measure

When the graph has fixed out-degree $\kappa$, then as observed before,

$$\mathbf{W} = \frac{1}{\kappa} \mathbf{A}.$$

Hence $\kappa \mathbf{W}$ is the adjacency matrix $\mathbf{A}$ for the graph. Mixing time measures how close $\mathbf{W}^t \mathbf{e}_i$ is to the stationary distribution. In other words, it measures the size of $\mathbf{W}_{n-1,i}^t$. This is $1/\kappa^t$ times the $a_{n-1,i}$ entry of $\mathbf{A}^t$. Now this entry of $\mathbf{A}^t$ counts the *number of oriented paths* of length $t$ in $\mathcal{G}$ from vertex $i$ to the terminal vertex. So, we have the conjectured equivalence: mixing time is small (e.g. $O(\log n)$) if and only if there are 'lots' of paths from 'most' vertices to $\tau$. In fact, we can formally prove the following proposition:

**Proposition 4.2.** *Suppose that the outdegree of every vertex other than $\tau$ is at least 2. Let $t$ be the average mixing time. Then for some $s \leq t$, the average number of paths from vertex $i$ to $\tau$ with length $s$ is at least $(3/4t)2^s$, where the average is taken over all vertices $i$ between 0 and $n - 1$.*

One way to interpret this is for the mixing time of $\mathcal{G}$ to be 'small', then there must be exponentially many paths from a typical vertex to $\tau$ with length less than the mixing time. We prove this statement in Appendix A. As having many paths seems useful for efficient data propagation, mixing time seems a good measure of how efficient our network will be under this computational graph.

Typically, we will treat mixing time as a *cutoff*, determining which graphs would require too many layers—much like the *under-reaching problem* (Barceló et al., 2020). It is also interesting to note that the fully connected graph does not have optimal mixing time—in fact, the additional paths create a *distraction*, and the mixing time is optimised by the "feedforward star" graph with edges $\mathcal{E} = \{(i, \tau) \mid i \in \mathcal{V}\}$. This relates to an observation by Di Giovanni et al. (2024), showing how fully-connected *undirected* graphs do not have optimal commute time due to distracting additional edges. However, we will not favour such solutions due to a lack of self-similarity.

## 4.5. Discussion About the Spectrum

In the usual theory of random walks on strongly connected graphs, mixing time is related to $\lambda_1$. This is the maximal eigenvalue of the normalised adjacency matrix (other than the eigenvalue 1).

However, in our case, there is *not* an obvious interpretation of mixing time in terms of the spectrum of $\mathbf{W}$. This is because $\mathbf{W}$ is lower-triangular. Hence, its spectrum is equal to its diagonal entries. Hence, as a set, the spectrum is just

$$\{1/\delta_{i\rightarrow} : 0 \leq i \leq n-1\}.$$

In general, these matrices are not diagonalisable, and so the total multiplicity of these eigenvalues may be less than $n$. For example, suppose that $\mathcal{G}$ has constant out-degree $\kappa$, apart from the terminal vertex. Then $\mathbf{W}$ just has two eigenvalues: 1 and $1/\kappa$.

# 5. Minimax Fidelity: Information Sharpness

Mixing time provides an excellent estimate of how long will information need to travel in a graph—it is generally a good idea to keep it low. However, it is not the whole story: it is irrelevant if information from a given node travels fast to the sink if only a small proportion of it makes it through. To quantify the extent to which information sharply reaches the sink, we will be using the *minimax fidelity* metric.

Additionally, we require that every node must have a positive in-degree; that is, $\delta_{i\leftarrow} > 0$ for all $i \in \mathcal{V}$. Note that this will be guaranteed by the self-edge property.

We want to track how "pre-disposed" this graph is to allowing information to travel freely in it. This relates to the over-squashing theorem in Barbero et al. (2024), but unlike them, we do not assume *any* degree of sharpness in choosing how information travels: specifically, at every step, we assume each node intakes the *average* value of all of the nodes over its incoming edges.

## 5.1. Fidelity

Starting from our adjacency matrix as before, we now derive a "diffusion" process specified by the following matrix $\boldsymbol{\Delta}$:

$$\Delta_{ij} = \begin{cases} 1/\delta_{i\leftarrow} & (j,i) \in \mathcal{E} \\ 0 & (j,i) \notin \mathcal{E} \end{cases} \tag{2}$$

Note that this is a *complementary* computation to the mixing time metric: while one normalises by *row*, the other normalises by *column*. And, further, we are interested in how sharply represented can a particular node be in the diffused representations, especially ones that reach the sink vertex.

To simulate this, we start with an idealised setting where all the mass concentrates in a specific vertex, $i$. The vector representing its weight may be expressed as $\mathbf{e}_i \in \mathbb{R}^n$, which is a one-hot vector that is one in position $i$ and zero elsewhere.

One step of diffusion corresponds to a matrix multiplication by $\boldsymbol{\Delta}$:

$$(\boldsymbol{\Delta}\mathbf{x})_j = \sum_j \Delta_{ij} x_j \tag{3}$$

(NB: this does not always yield a probability distribution!)

We can simultaneously estimate the diffusion properties for all $n$ possible initial vertices by stacking the $\mathbf{e}_i$ vectors, recovering the identity matrix, $\mathbf{I}$. Accordingly, after $t \geq 0$ layers of propagation, we can read off the coefficients of each item in each receiver by computing $\boldsymbol{\Delta}^t$.

Since we take particular care on how much information has reached *the sink vertex*, we can define the ***fidelity*** *of node $i$ at $t$ steps* as $\boldsymbol{\Delta}_{\tau i}^t$. This can be interpreted as: "what is the coefficient of node $i$ in the weighted sum within $\tau$ after $t$ steps of averaging diffusion?"

An important point about the fidelity measure is that it does not always grow with increasing $t$—this is easy to observe in nodes near the end of the sequence:

**Proposition 5.1.** *Let $\mathcal{G} = (\mathbb{Z}_n, \mathcal{E})$ be a feedforward graph of $n > 1$ nodes with all self-edges and a unique sink vertex $\tau = n-1$; that is, every node $i \in \mathbb{Z}_n$ is connected to $\tau$ by a path in $\mathcal{E}$. Then, if $\delta_{(n-2)\leftarrow} > 1$, as $t \rightarrow \infty$, $\boldsymbol{\Delta}_{\tau,n-2}^t \rightarrow 0$, that is, the fidelity of node adjacent to the sink eventually vanishes if it has at least one nontrivial incoming edge.*

*Proof.* Initially, $\boldsymbol{\Delta}_{n-2,n-2}^0 = 1$, and $\boldsymbol{\Delta}_{j,n-2}^0 = 0$ for all $j \neq n-2$. Since information can only travel forwards in the graph, we can conclude that, if $j < i$, $\boldsymbol{\Delta}_{ji}^t = 0$ for all $t$. Hence, it will be sufficient to track $\boldsymbol{\Delta}_{n-2,n-2}^t$ and $\boldsymbol{\Delta}_{\tau,n-2}^t$ over time for the purpose of this proof.

Following the formula, we conclude $\boldsymbol{\Delta}_{n-2,n-2}^{t+1} = \Delta_{n-2,n-2}\boldsymbol{\Delta}_{n-2,n-2}^t = (\delta_{(n-2)\leftarrow})^{-1}\boldsymbol{\Delta}_{n-2,n-2}^t = (\delta_{(n-2)\leftarrow})^{-t}$. Since we assumed $\delta_{(n-2)\leftarrow} > 1$, this value will certainly decay towards zero as $t \rightarrow \infty$.

Since there is only one sink vertex, $\tau$, it must have an in-degree $\delta_{\tau\leftarrow} > 1$: it must have a self-edge (by assumption) and it must have a direct incoming connection from at least one other preceding node. Further, at least one of those nodes must be $n-2$, otherwise it would introduce another sink. The fidelity of node $n-2$ at $t+1$ layers can hence be expressed as $\boldsymbol{\Delta}_{\tau,n-2}^{t+1} = \Delta_{\tau,n-2}\boldsymbol{\Delta}_{n-2,n-2}^t + \Delta_{\tau\tau}\boldsymbol{\Delta}_{\tau,n-2}^t = (\delta_{\tau\leftarrow})^{-1}\left((\delta_{(n-2)\leftarrow})^{-t} + \boldsymbol{\Delta}_{\tau,n-2}^t\right)$.

Now, note that this expression is maximal when $\delta_{\tau\leftarrow} = \delta_{(n-2)\leftarrow} = 2$. As such, we can bound $\boldsymbol{\Delta}_{\tau,n-2}^{t+1} \leq 2^{-t-1} + \boldsymbol{\Delta}_{\tau,n-2}^t/2$. From this, we can prove by induction that $\boldsymbol{\Delta}_{\tau,n-2}^t \leq t/2^t$. The base case ($t = 0$) clearly holds, as $\boldsymbol{\Delta}_{\tau,n-2}^0 = 0 \leq 0$. From there, assuming the bound

holds for $t$, we can show it holds for $t+1$ by substituting $\boldsymbol{\Delta}_{\tau,n-2}^{t+1} \leq 2^{-t-1} + (t/2^t)/2 = (t+1)/2^{t+1}$.

This upper bound on $\boldsymbol{\Delta}_{\tau,n-2}^t$ also decays towards zero as $t \to \infty$, settling the proof. □

## 5.2. (Mini)max Fidelity

Given this result, it does not make sense to study long-term dynamics of fidelity for most nodes. Rather, we can track the "best-case" scenario for node $i$, as the *maximal* fidelity it achieves over the lifetime of the diffusion; that is, $\phi_i = \max_t \boldsymbol{\Delta}_{\tau i}^t$. This is the most sharp we can ever extract node $i$'s features in the sink node[2].

Out of all of these sharpness measures, we are hunting for the "weakest points", where the model is particularly vulnerable to extracting a particular node's features, at any layer. As such, we compute the **minimax fidelity** as

$$\min_i \phi_i = \min_i \max_t \boldsymbol{\Delta}_{\tau i}^t \qquad (4)$$

We will now deploy this measure on some graph generators of interest, just as we did for mixing time.

It is worth noting that tracking how the signal evolves over time in this way is similar to the Dirichlet energy, which has been used extensively to study the over-smoothing effect in GNNs (Cai & Wang, 2020; Zhou et al., 2021; Rusch et al., 2022; Di Giovanni et al., 2022). One key difference is in the fact that the measure at hand here involves taking maxima and minima and hence is not as smooth to work with.

### 5.2.1. FULLY CONNECTED GRAPHS (AND NORMALISED MINIMAX FIDELITY)

While fully connected graphs had highly favourable logarithmic mixing time, they immediately fully average information, and therefore their minimax fidelity is always $1/n$. This is potentially problematic, especially if hit with dispersion issues at inference time (Veličković et al., 2024).

Taking this into account, and given the ubiquitous use of the fully connected feedforward graphs in contemporary architectures, we will often opt to report the *normalised minimax fidelity*:

$$n \min_i \phi_i = n \min_i \max_t \boldsymbol{\Delta}_{\tau i}^t \qquad (5)$$

which ensures that the fidelity of the fully connected graph is always 1 (see Figure 1), and provides an intuitive threshold for whether the fidelity is more or less favourable than fully connected graphs.

### 5.2.2. LINE GRAPHS

What line graphs sacrifice in mixing time (and hence tractability of certain problems) they make up for in fidelity. Indeed, line graphs are among the graph families with highest minimax fidelity. Their minimax fidelity can be expressed as follows[3], for a graph of $n$ elements:

$$\max_a \frac{\binom{a}{n-1}}{2^a} \qquad (6)$$

This quantity also decays to zero as $n \to \infty$ (see Appendix B for a proof using Stirling's approximation), but its normalised minimax fidelity consistently grows as $\sqrt{n/\pi}$ (see Figure 1—indeed, line graphs have the highest initial fidelity of all considered graphs).

### 5.2.3. ERDŐS-RÉNYI GRAPHS AND ORIENTED EXPANDERS

So far, we have mainly examined line graphs and fully connected graphs – which in ways correspond to two extremes with respect to the metrics we proposed.

We may further see from Figure 1 that creating oriented versions of Erdős-Rényi (Erdős et al., 1960) graphs (keeping indegrees fixed) as well as orienting graphs that are known to be undirected expanders (by assigning a random permutation to their nodes) results in negative performance on *both* of our metrics compared to the fully connected graph.

We will not study these graphs in more depth here, though it is worth remarking that fixed-indegree Erdős-Rényi graphs are very likely to introduce additional sinks, breaking one of our theory's key assumptions. Another reason why both of these distributions are likely to fail is that they are *effectively unbiased* in terms of which incoming edges they sample – an earlier node is equally likely to be assigned as an indegree neighbour as a latter one. As we've seen in Proposition 5.1, latter nodes are particularly vulnerable to losing fidelity quickly. This motivates our next distribution under study.

### 5.2.4. POISSON($p$) GRAPHS

Guided by this observation, we set out to construct a graph generator which is *biased* to create edges towards the end of the sequence (so it preserves fidelity better) while still having a chance of generating an edge which is further away from the target node (so it shortens mixing time).

This combination of constraints led us to consider a graph which samples edges by going right-to-left, simulating a Poisson process with probability $p$. It samples in-degree neighbours of node $i$ (given an indegree budget) as follows:

$$\mathcal{E} \leftarrow \mathcal{E} \cup \{(i, i)\}$$

---

[2]It is assumed implicitly that neural networks deeper than optimal may still be able to preserve this level of information of node $i$ by leveraging their residual connections (He et al., 2016).

[3]Diffusing a 1 along the line graph basically involves generating normalised entries of a (trimmed) Pascal's triangle, which is why the binomial coefficients appear in this formula.

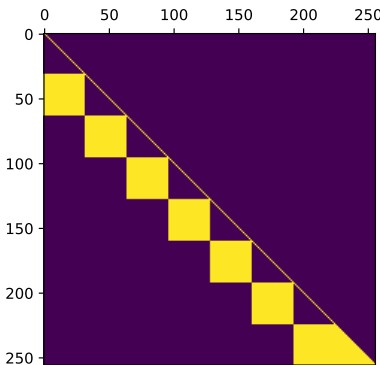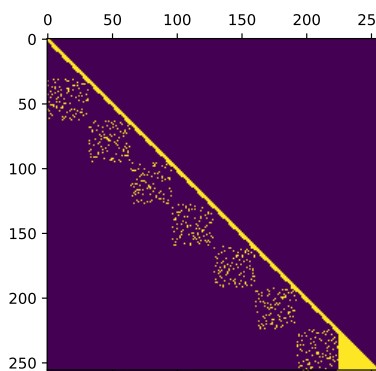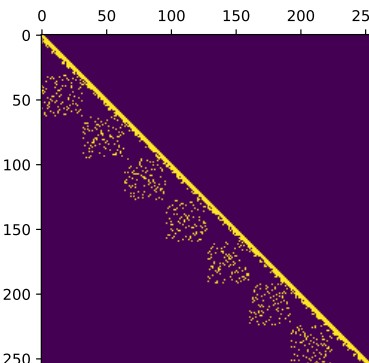

*Figure 2.* The evolution of the adjacency matrices generated by the FS graph generator. (**Left:**) The original solution found by FunSearch, when asked to optimise minimax fidelity while keeping the mixing time under $2 \log_2 n$. (**Middle:**) Adapting the initial solution to bound the in-degree within each cluster. Each pair of successive blocks are now connected by a random bipartite expander. (**Right:**) The final FS graph generator, leveraging a recursive construction to ensure self-similarity and sufficient sparsity.

```
edges ← 1
j ← i − 1
while j ≥ 0 ∧ edges < budget do
    ρ ∼ U(0, 1)
    if ρ > p then
        E ← E ∪ {(j, i)}
        edges ← edges + 1
    end if
    j ← j − 1
end while
```

Different values of $p$ will yield graphs with different levels of locality ($p = 0$ yields a local graph generator). In our experiments, we found that $p = 0.2$ struck the best balance between improving fidelity and penalising mixing time, though the mixing time is still not nearly as controllable as some of the other graphs – the profile of the metrics achieved by $\mathrm{Poisson}(0.2)$ is provided in Figure 1.

## 6. The FunSearch (FS) Graph Generator

Apparently, even after trying out many hand-crafted sparse graph generators, we have not been able to strike the right balance between mixing time and fidelity. So we turned our attention to *evolutionary methods*: as there's no clear method of constructing graphs which optimise the fidelity while keeping the mixing time within reasonable bounds, we used the FunSearch (Romera-Paredes et al., 2023) algorithm to produce graphs with good values for both metrics. The most promising result generated by FunSearch may be found in Figure 2 (Left), and it turns out to be an impactful motif.

The core idea of this graph is to divide the nodes into $O(\log n)$ groups, then applying full bipartite graphs across successive chunks. Since there aren't almost any intra-cluster edges, random walks almost certainly always move from one cluster to the next, mixing in logarithmic time. And since each chunk only attends over a smaller number of nodes $O(n/\log n)$, this improves fidelity.

There are two core issues with this graph left to fix from an asymptotics point of view. First, the indegree of each node is $O(n/\log n)$ as $n \to \infty$, which is undesirable. Second, the graph does not exhibit self-similarity: it will only work well if the predictions will take place in the final node block, which will be exposed to the "final triangle" of full edges.

In order to fix these issues, we first note that each filled chunk of the matrix is essentially corresponding to edges of a bipartite graph. And constructions of bipartite expander graphs (which have excellent neighbourhood coverage) are well understood in mathematics. In Figure 2 (Middle), we provide one such construction, built by concatenating $\kappa$ random perfect matchings (Lubotzky, 1994; Sarnak, 1990).

To fix self-similarity, it is evident that all intra-cluster edges (the "blank triangles" in the adjacency matrix) need to be filled equivalently. However, this would result in $O(n)$ asymptotic indegree per node. As such, we opt to fill the intra-cluster edges *recursively*, by constructing smaller copies of the FS graph, within each triangle. By doing so, we recover the graph from Figure 2 (Right), which is now both suitably sparse and self-similar!

As can be seen in Figure 1, the FS graph generator indeed has a desirable tradeoff between solid normalised minimax fidelity and contained mixing time. Further, the fidelity can be meaningfully controlled by varying the chosen indegree of each expander graph; for example, out of all choices of

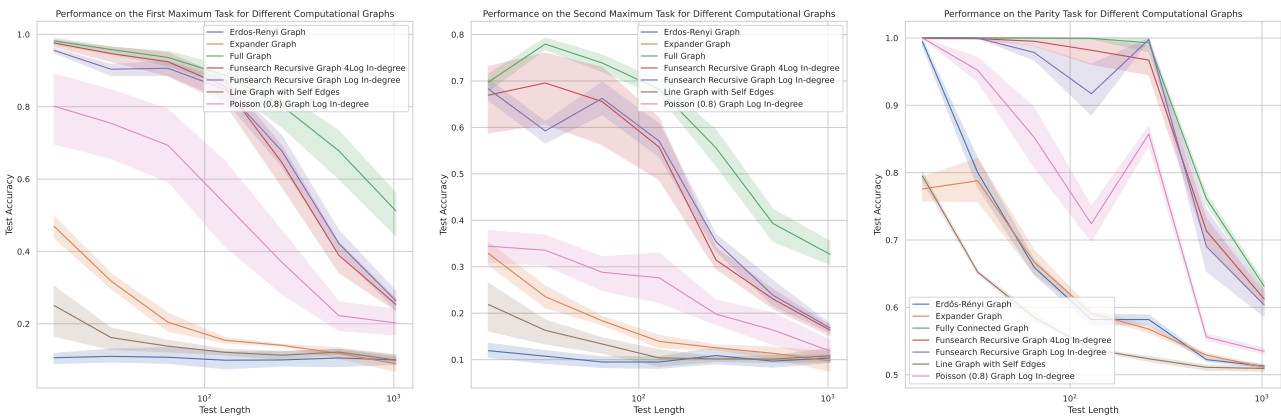

*Figure 3.* Performance profiles of GATs trained with various computational graphs, on the max retrieval task (**left**), second max retrieval task (**middle**), and parity task (**right**), when trained up to size 256. The $x$-axis corresponds to test accuracy on sizes from 16 to $1,024$.

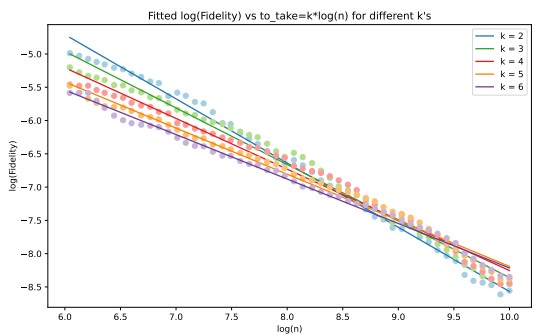

*Figure 4.* Fitted scaling laws of log-minimax fidelity for different indegrees (of the form $k \log n$ for different values of $k$ over $n$ nodes) of sampled expander graphs in the FS graph.

$O(\log n)$ indegree, we find that $4 \log n$ offers a favourable "fidelity scaling law" (see Figure 4).

### 6.1. Analysing the Mixing Time of the FS Graph

Having demonstrated that the FS graph generator has high fidelity and low mixing time in an empirical setting, we will now prove one of these properties—specifically, that its mixing time is favourable.

Informally, the idea of the proof is that the partitioning of the nodes into $O(\log n)$ blocks allows for quickly traversing long distances in the graph, via the bipartite edges. Making this construction recursive means that, once we reach the final block of size $O(n/\log n)$, the progression towards the sink will remain fast within that block, avoiding a linear mixing time as $n \to \infty$.

However, we also need to take care that the recursively introduced edges (which stay in the same block) do not overly impede the random walker's trajectory across blocks. Formally, we prove the following:

**Theorem 6.1.** *Consider a FS graph generator with $\lceil \log n \rceil$ blocks per level for a graph of $n$ nodes. Further, assume that for every node, the proportion of its outgoing edges going across blocks is lower bounded by a constant $\alpha > 0$. Then the mixing time of graphs produced by this generator is $O(\mathrm{polylog}\, n)$.*

For reasons of brevity, we provide a full proof of this Theorem (along with supplementary remarks concerning how to construct the graph to match the assumptions of the Theorem) in Appendix C.

## 7. Results

To supplement our findings with empirical performance metrics, we evaluate a graph attention network (Veličković et al., 2018)-style model on three representative tasks:

- finding the highest valued node in a sequence (Veličković et al., 2024);

- finding the second highest valued node in a sequence (Ong & Veličković, 2022);

- computing the parity value of a bitstring (Hahn, 2020).

using adjacency matrices sampled from our graph generators. The tasks are chosen to represent opposite extremes of sharpness required: for finding the maximum, exactly one node needs to propagate its input features to the sink – for parity, *all* of them need to. The performances of different computational graphs can be seen in Figure 3.

'(Second) Maximum retrieval' tasks consist in finding the category of the (second) maximum element in the set of items. Categories are provided one-hot encoded, together with the values. For each set the categories are consistent, but are randomised between different sets. Ten categories are used, so chance accuracy is $\sim 10\%$.

The 'Parity' task consists of computing the parity of the number of 1s in a bitstring. Since there are only two possible parities, odd and even, random guessing should lead to around $50\%$ accuracy on the task.

For details on the models' architectures and optimisation, please see Appendix E. As one can see in Figure 3, the FS graph leads in general to the best performance in distribution and best out of distribution generalisation when compared to other sparse graphs—matching the intuition found by our metrics. More strongly, in the Parity task the FS graph manages to effectively match the fully connected graph, while having a substantially lower number of edges, and while its performance tends to be similar to the Poisson(0.8) graph inthe Maximum task, in the Second Maximum and Parity tasks there's a significant gap in performance between the FunSearch graph and all the other sparse graphs.

Additionally, we may note that the results are aligned with the recommendations given in Figure 4, which concerns the problem sizes up to $1,024$ studied here. Namely, leveraging an in-degree of $4 \log n$ for the random expanders used to sparsify the FS graph has a more graceful performance profile with increasing problem size across all three tasks compared to using an in-degree of $\log n$.

**On the Importance of Self-Edges and In-Degrees**   Our theoretical framework always assumes that all *self-edges* are included, as a valuable way to preserve each node's information over time. Accordingly, removing it led to significant performance regressions on all tasks – especially on the Parity task where multiple elements need to interact meaningfully for the final answer computation.

We also evaluated our graph generators at different orders-of-magnitude of in-degree – from $O(1)$ to $O(\log n)$ to $O(\sqrt{n})$. We find that, as the in-degrees are increased, the models get more performant in-distribution, without a significant effect on overall out-of-distribution performance.

## 8. Conclusions

In this work, we have embarked on a detailed study of feedforward computational graphs, attempting to chart a novel path that could enable for a principled discovery of useful feedforward blueprints. While it is apparent that analysing these graphs is substantially less straightforward than doing so with their undirected counterparts, we believe the outcomes to have been fruitful. Namely, we proposed two

well-justified metrics of feedforward information propagation, used them to automatically discover an interesting graph generator, and demonstrated its strong performance on several carefully crafted benchmarks.

We are hopeful that our work will inspire targeted follow-up studies in this space, which may usher in a new paradigm of graph rewiring. To this end, we particularly believe it would be worthwhile to study computational graphs that *dynamically change* across layers – as opposed to a single fixed graph studied here – as well as attempting to establish a more formal framework connecting mixing time, fidelity and similar metrics. The framework of Lim et al. (2024) could be useful for reasoning about feedforward computational graphs spanning multiple layers.

## Acknowledgements

We would like to extend deep thanks to Steven Kapturowski for detailed assistance with the parity experiment, as well as Alexander Novikov for kindly supporting us as we were setting up our FunSearch experiments. We are also grateful to Razvan Pascanu and Simon Osindero for diligently reviewing this paper and providing valuable comments. Marc Lackenby was partially supported by the Engineering and Physical Sciences Research Council (grant number EP/Y004256/1).

## Impact Statement

This paper presents a novel framework for improving the information propagation quality across data with a feedforward constraint. Such models are highly prevalent, and innovations in this space could eventually lead to improvements in model sparsity (allowing for serving more powerful models on edge devices, for example) and model out-of-distribution generalisation (alowing more capable reasoning and scientific advances). Any societal consequences of our work could hence be equated to the societal consequences of accelerating these flavours of research.

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

## A. Proof of Proposition 4.2.

*Proof.* For $0 \leq i < n - 1$, let $p(i, s)$ be the number of paths from vertex $i$ to $\tau$ with length $s$. Then

$$(\mathbf{W}^t)_{\tau i} \leq \sum_{s \leq t} \frac{p(i, s)}{2^s}.$$

This is because $(\mathbf{W}^t)_{\tau i}$ represents the probability of a random walk starting at $i$ and reaching $\tau$ by time $t$. This is equal to the sum, over all $s \leq t$, of the probability that the random walk first reaches $\tau$ at time $s$. For any such $s$, this probability is equal to the sum, over all paths from $i$ to $\tau$ with length $s$ avoiding the loop based at $\tau$, of the probability of taking that path. This latter probability is at most $1/2^s$ because at each step along the path there were at least two options that could have been taken.

For $t$ equal to the averaged mixing time,

$$\frac{1}{n} \sum_{i=0}^{n-1} (\mathbf{W}^t)_{\tau i} \geq \frac{3}{4}.$$

So,

$$\sum_{s \leq t} \frac{1}{n} \sum_{i=0}^{n-1} \frac{p(i, s)}{2^s} = \frac{1}{n} \sum_{i=0}^{n-1} \sum_{s \leq t} \frac{p(i, s)}{2^s} \geq \frac{1}{n} \sum_{i=0}^{n-1} (\mathbf{W}^t)_{\tau i} \geq \frac{3}{4}.$$

Therefore, for some $s \leq t$,

$$\frac{1}{n} \sum_{i=0}^{n-1} \frac{p(i, s)}{2^s} \geq \frac{3}{4t}.$$

So, the average number of paths from vertex $i$ to $\tau$ with length $s$ is at least $(3/4t)2^s$. $\qquad\square$

## B. Proof that Minimax Fidelity in a Line Graph Decays to Zero with Increasing Size

First let's determine an expression for $\max_a \frac{\binom{a}{n-1}}{2^a}$ for a fixed $n$.

Notice that this is equivalent to finding an expression for $\max_a \frac{\binom{a}{k}}{2^a}$ and then replacing $k = n - 1$.

With a fixed $n$ we know that $a \geq k$, otherwise the binomial term would simply be 0. Now let $f(a) = \frac{\binom{a}{k}}{2^a}$, and let's compute $\frac{f(a+1)}{f(a)}$:

$$\frac{f(a+1)}{f(a)} = \frac{2^a}{2^{a+1}} \frac{\binom{a+1}{k}}{\binom{a}{k}}$$

$$\frac{f(a+1)}{f(a)} = \frac{1}{2} \frac{(a+1)!}{k!(a-k+1)!} \frac{k!(a-k)!}{a!}$$

$$= \frac{1}{2} \frac{(a+1)}{a-k+1}$$

From that we know that $\frac{f(a+1)}{f(a)} \geq 1 \iff (a+1) \geq 2(a - k + 1) \iff 2k - 1 \geq a$. As such $f(a)$ increases for a in $[k, 2k-1]$ and decreases for $a > 2k$. We can thus say that

$$\max_a \frac{\binom{a}{k}}{2^a} = \frac{\binom{2k}{k}}{2^{2k}}$$

Now let's compute our limit using Stirling's Approximation:

$$\lim_{k\to\infty} \frac{\binom{2k}{k}}{2^{2k}} = \lim_{k\to\infty} \frac{(2k)!}{k!^2} \cdot \frac{1}{2^{2k}} =$$

$$\lim_{k\to\infty} \frac{(2k)^{2k}}{e^{2k}} \sqrt{2\pi(2k)} \frac{e^{2k}}{k^{2k} \cdot (2\pi k)} \cdot \frac{1}{2^{2k}} =$$

$$\lim_{k\to\infty} 2^{2k} \sqrt{\frac{1}{\pi k}} 2^{-2k} = 0$$

## C. Proof of Theorem 6.1.

*Proof.* In order to help us reason about the mixing time, it will be very useful to compute it as a function of the starting level and block the random walker is in. That is, we will track $T(d, k)$ as the expected number of steps needed for a random walker to reach the sink node, assuming the walker is currently $d$ depth levels away from the deepest recursive level, and $k$ blocks away from reaching the final block in the current depth level.

We already know that $k$'s maximal value will be $\lceil \log n \rceil$ for each depth level, due to the generator's parameters. For the maximal value of $d$, i.e. the total number of depth levels, we note that after each level, the number of nodes being considered is further subdivided by $\lceil \log n \rceil$. This means that, after $l$ levels, the block size is $\frac{n}{\lceil \log n \rceil^l}$. No further subdivisions are possible once the block size reaches $1$.

Denoting the total number of levels by $D$, we have that

$$n/(\lceil \log n \rceil)^{D+1} < 1, \qquad n/(\lceil \log n \rceil)^D \geq 1$$

and hence

$$\log n < (D + 1) \log\lceil \log n \rceil, \qquad \log n \geq D \log\lceil \log n \rceil.$$

So

$$D = \left\lfloor \frac{\log n}{\log\lceil \log n \rceil} \right\rfloor.$$

With this in mind, we aim to quantify $T\left(\left\lfloor \frac{\log n}{\log\lceil \log n \rceil} \right\rfloor, \lceil \log n \rceil\right)$. To do this, we will establish several upper bounds on $T(d, k)$, assuming pessimistic behaviour from the random walker.

Firstly, when the walker is in the final level ($d = 0$), the number of nodes considered is $1$, hence there are are no blocks left to traverse, and therefore $T(0, k) = 0$ for all $k$.

Then, once the walker hits the final block in its current depth level ($k = 0$), it automatically transitions into the next level, which is further subdivided into $\lceil \log n \rceil$ blocks. In the worst-case scenario, the walker will have landed in the very first block of the next depth level, and will need to traverse them all. Hence, $T(d, 0) \leq T(d - 1, \lceil \log n \rceil)$ for all $d > 0$.

Finally, in all other cases, we leverage the assumption on the outgoing edge ratio to remark that the walker will transition into the next block at its current depth level with probability at least $\alpha$. We make a pessimistic assumption that, if the walker does not transition into the next block, it certainly stays exactly put within its current block. This leads to the upper bound of $T(d, k) \leq 1 + \alpha T(d, k - 1) + (1 - \alpha)T(d, k)$.

We now define $\tilde{T}(d, k)$ recursively using the following recurrence relations:

$$\tilde{T}(d, k) = 1 + \alpha\tilde{T}(d, k - 1) + (1 - \alpha)\tilde{T}(d, k) \tag{7}$$

$$\tilde{T}(d, 0) = \tilde{T}(d - 1, \lceil \log n \rceil) \tag{8}$$

$$\tilde{T}(0, k) = 0 \tag{9}$$

A simple induction gives that $T(d, k) \leq \tilde{T}(d, k)$. So it suffices to find an upper bound for $\tilde{T}(d, k)$.

The key observation is, since $\alpha$ is assumed identical in every depth level, and $k$ is always reset to $\lceil \log n \rceil$ after each depth level, we can equivalently represent the total walk time as a direct sum of walk times on each depth level.

Then, since each depth level is represented as a chain of $\lceil \log n \rceil$ states, $\tilde{T}(d, k)$ may be observed as the expected time of traversing a line graph of $k + (d - 1)\lceil \log n \rceil$ steps and transition probability $\alpha$. The total number of such steps is $D\lceil \log n \rceil$, for $D$ being the number of depth levels.

To compute the mixing time of such a Markov chain, we assume that the walker starts in the first node of the first block, and we want to know the smallest value of $t$ such that

$$\sum_{j \leq D\lceil \log n \rceil} \binom{t}{j} \alpha^j (1 - \alpha)^{t-j} < 1/4.$$

For ease of notation, set $N = D\lceil \log n \rceil$. The left hand side is the cumulative distribution function for the binomial random variable $X(t, \alpha)$. So we must find the smallest value of $t$ such that $\mathbb{P}(X(t, \alpha) \leq N) < 1/4$. Now, Hoeffding's inequality gives that

$$\mathbb{P}(X(t, \alpha) \leq N) < \exp\left(-2t\left(\alpha - \frac{N}{t}\right)^2\right).$$

Hence, when $t = 2N/\alpha$,

$$\mathbb{P}(X(t, \alpha) \leq N) < \exp\left(-N\alpha\right).$$

When $\alpha$ is fixed and $N$ is large, this is certainly less than $1/4$. This establishes that the mixing time is at most $O(N) = O(D \log n)$. Note that when $t = D\lceil \log n \rceil$, then

$$\sum_{j \leq D\lceil \log n \rceil} \binom{t}{j} \alpha^j (1 - \alpha)^{t-j} = 1.$$

So the mixing time is certainly at least $D\lceil \log n \rceil$.

Therefore, plugging in the appropriate values of $D$, we can conclude that the final mixing time will be $O\left(\frac{\log^2 n}{\log \log n}\right)$, which is $O(\text{polylog}\, n)$, completing the proof. $\qquad\square$

*Remark* C.1. An important assumption for our mixing time derivation is that the outdegree ratio of each node's cross-block outgoing edges can be lower bounded by a constant (which we can use as the value of $\alpha$ for our analysis).

Since there will be approximately $\frac{\log n}{\log \log n}$ depth levels in total, we cannot maintain a fixed outdegree of $\delta_{i\rightarrow}$ at each depth level, as the ratio is then $\frac{\delta_{i\rightarrow}}{\delta_{i\rightarrow}\frac{\log n}{\log \log n}} = \frac{\log \log n}{\log n}$, which will decay to zero as $n \rightarrow \infty$ and cannot be lower-bounded by a constant. This implies that the relative out-degree of each node in each block must decay with increasing depth level.

One simple way to support such a decay is to assume a *geometric* decay with ratio $r < 1$; that is, if the amount of cross-block edges from a given node, $i$, at a given depth level, $d$, is $\delta_{i\rightarrow}^d$, the amount of cross-block edges from that node at the next depth level would be $\delta_{i\rightarrow}^{d-1} = \delta_{i\rightarrow}^d r$.

This process upper-bounds the total number of outgoing edges of every node in the final graph by a geometric series $\delta_{i\rightarrow}^d + \delta_{i\rightarrow}^d r + \delta_{i\rightarrow}^d r^2 + \cdots = \frac{\delta_{i\rightarrow}^d}{1-r}$. As such, the proportion of the edges which are cross-block at the current level would correspond to $\frac{\delta_{i\rightarrow}^d}{\frac{\delta_{i\rightarrow}^d}{1-r}} = 1 - r$. When moving to the next depth level, the number of outgoing edges will be $\delta_{i\rightarrow}^d r$, and since the previous level has already been fully crossed, there will be no edges from the previous level, and the total outdegree is $\frac{\delta_{i\rightarrow}^d r}{1-r}$, also leading to a $1 - r$ ratio. This trend continues with increasing depth level, and we can hence use $1 - r$ as our pessimistic estimate of $\alpha$.

## D. Adjacency Matrix Gallery

In Figures 5–7 we provide visualisations of several adjacency matrix samples for a variety of graph generators we studied in this paper.

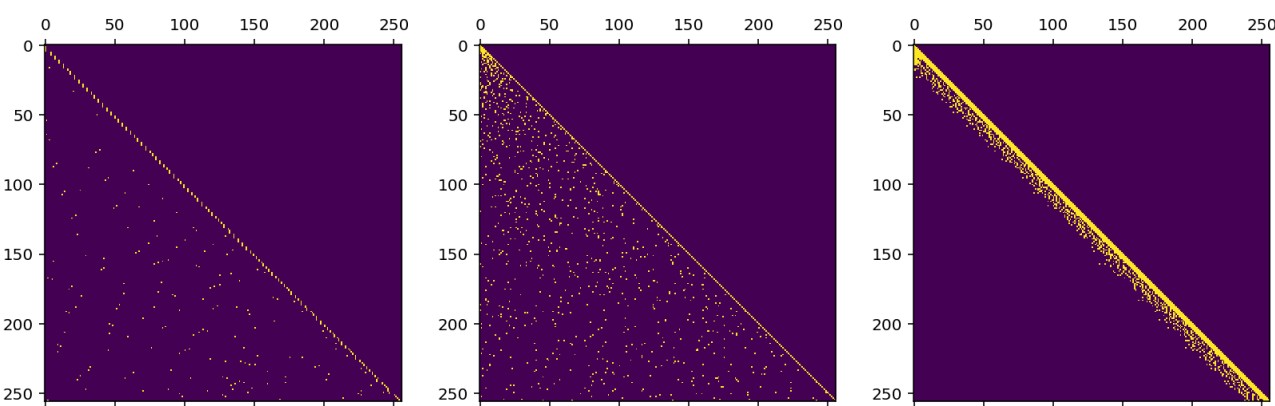

*Figure 5.* Generated samples of adjacency matrices for the graph generators of Oriented Expander (**left**), Erdős-Rényi with constant indegree (**middle**) and Poisson(0.2) with constant indegree (**right**).

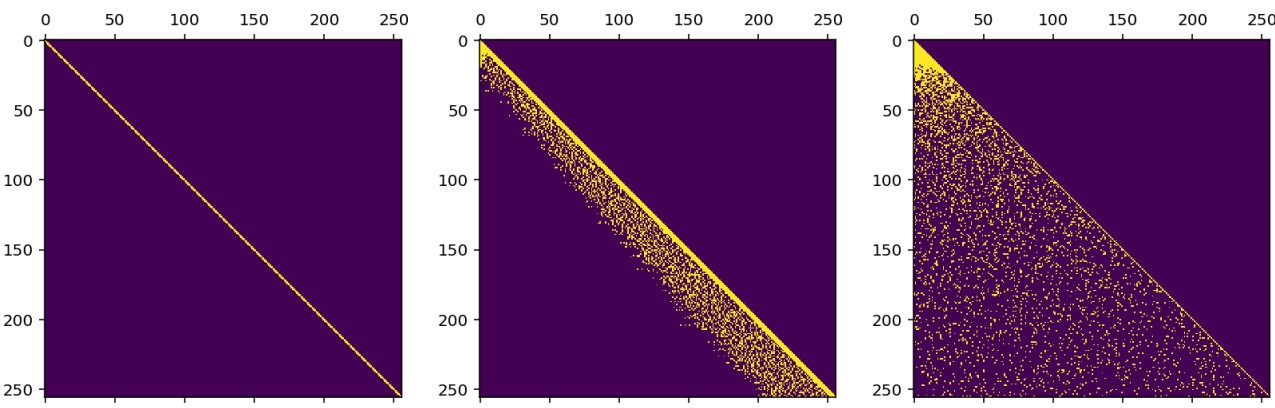

*Figure 6.* Generated samples of adjacency matrices for the graph generators of Line Graphs (**left**), Poisson(0.2) with $\sqrt{n}$ indegree (**middle**) and Erdős-Rényi with $\sqrt{n}$ indegree (**right**).

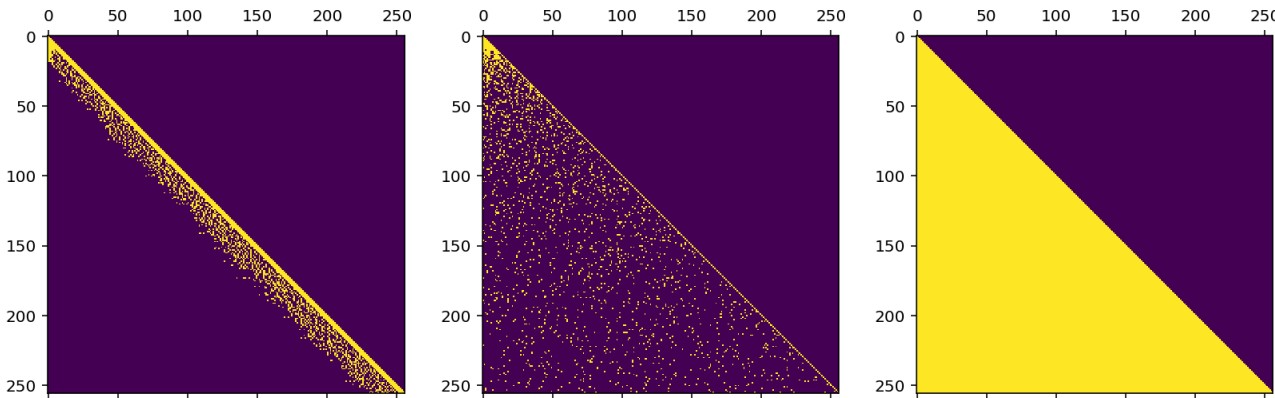

*Figure 7.* Generated samples of adjacency matrices for the graph generators of Poisson(0.2) with $\log_2 n$ indegree (**left**), Erdős-Rényi with $\log_2 n$ indegree (**middle**) and the Fully Connected graph (**right**).

# E. Models and Training

We test graphs on two different tasks: bit set parity, which is well known to be hard for transformer architectures without Chain Of Thought (Kim & Suzuki, 2024) and finding the category of the maximum or second maximum element in the set. We are interested in length generalisation properties of the tasks. So while we train on length up to 256 elements, we test sequences of up to $1,024$ elements. We tuned the order-of-magnitude of the learning rate and the weight decay coefficient using the fully-connected graph experiments only, and then reused the tuned parameters everywhere else.

## E.1. Maximum Retrieval

For 'maximum' tasks we utilise simple transformer architecture. It consists of one attention block, which is a single attention layer followed by two feedforward layers with GELU activations and pre-norm in between. This attention block is repeated (with shared parameters) for $\log n$ steps for an input of $n$ nodes. We use the cross-entropy loss function and the AdamW optimiser with $10^{-3}$ learning rate and 256 batch size. Training is performed over $10,000$ steps. We utilise casual attention mask in conjunction with mask provided by graphs.

## E.2. Parity

We use a model similar to Gemma 2 (Team et al., 2024b), but with only only 8 layers and using standard multi-head attention with 8 heads, the vocabulary size was 2, as there are only 2 symbols in a bitstring, and the embedding dimension was 256. The loss is a crossentropy and is computed over all positions in the sequence, so the model must predict the parity for all prefixes of the input bitstring. To train the model we used the LaProp optimizer (Ziyin et al., 2021) with weight decay and RMSClip (Shazeer & Stern, 2018) for 1 million steps, with batch size of 128 sequences. Our hyperparameters were:

*Table 1.* LaProp Hyperparameters

| Hyperparameter | Value |
| --- | --- |
| Learning Rate | $1 \times 10^{-3}$ |
| $\beta_1$ | 0.9 |
| $\beta_2$ | 0.9 |
| Weight Decay | $5 \times 10^{-4}$ |
| RMSClip's d | 1 |

As an indication of the utility of various feedforward graphs in the setting where nodes correspond to natural language tokens, we also fine-tuned Gemma 2B (Team et al., 2024a) – utilising these graphs as attention masks across all Transformer layers – on the standard Wikipedia dataset[4] containing texts obtained from Wikipedia database dumps.

After $3,000$ batches of training, we found that the perplexities obtained by the FS graph were comparable with the fully connected graph, while significantly improving on the line graph—a trend which is consistent throughout training.

---

[4]https://www.tensorflow.org/datasets/catalog/wikipedia

