# OpenReview forum: "What Makes a Good Feedforward Computational Graph?"
_ICML.cc/2025/Conference — ICML 2025 poster_

### Official Review · Reviewer_QmmN · 2025-02-17

**Overall Recommendation:** 3

**Summary:**

The authors are motivated by the recent surge of feedforward networks, and analyze the underlying computational graphs. Their core question is: What characterizes a “good” computational graph? To address this, they propose two metrics:
a) Mixing time: Assesses how quickly information from various nodes reaches a designated target node, and
b) Minimax fidelity: Evaluates whether information vanishes along the paths from input nodes to the target node (i.e., whether bottlenecks exist).
They study different random graph models under these metrics, focusing on their asymptotic behavior. They then train a GAT on graphs derived from these models to empirically illustrate some aspects of their theoretical insights.

**Claims And Evidence:**

- Although the paper is motivated by feedforward networks, after the initial setup, there is little tangible connection between these metrics and actual modern neural network architectures. The authors’ own references to empirical performance remain vague, for instance:
“As having many paths seems useful for efficient data propagation, mixing time seems a good measure...”
- Yet such intuition does not necessarily hold for widely used architectures like deep MLPs or Transformers, which can often be represented by combinations of bipartite graphs (leading to high mixing time and low minimax fidelity) yet still excel in performance.
- Much of the argument feels more opinion-based than backed by empirical evidence. For instance, low mixing time might not be inherently desirable. For example, deep ResNets or GPT architectures have ±96 layers to enable the learning of "strong" hidden representations without directly propagating information to output neurons.
- Similarly, a high minimal fidelity does not obviously correlate with effectiveness. Modern networks routinely prune away large portions of weights or nodes, indicating that focusing on relatively few “important” nodes/edges is advantageous.
- In contrast to their claims, the authors do not demonstrate any clear empirical link between these proposed metrics and the predictive performance of real-world neural networks.

**Essential References Not Discussed:**

The authors claim that "many important concepts have not been generalised to the directed case..." I would suggest [1] where concepts like oversmoothing etc. were generalized to directed graphs.

I would also recommend [2] that discuss how feedforward networks / their parameters can be written as feedforward graphs.

[1] Maskey, S., Paolino, R., Bacho, A., & Kutyniok, G. (2024). A fractional graph laplacian approach to oversmoothing. Advances in Neural Information Processing Systems, 36.

[2] Lim, D., Maron, H., Law, M. T., Lorraine, J., & Lucas, J. Graph Metanetworks for Processing Diverse Neural Architectures. In The Twelfth International Conference on Learning Representations.

**Experimental Designs Or Analyses:**

Experimental design and analysis appear sound.

**Methods And Evaluation Criteria:**

The datasets do not provide meaningful insights into what constitutes a “good” computational graph. They consist of synthetic graphs from the different random graph models, on which three different tasks are tested using a GAT. However, there is no clear connection to feedforward networks or their suitability for different tasks, leaving the practical relevance of these experiments unclear.

**Other Comments Or Suggestions:**

In conclusion, although the authors appear to address their central question, they do not provide sufficient evidence that these metrics correlate with the performance of modern feedforward networks. The authors’ motivation feels overstated: they neither fully analyze existing feedforward architectures nor demonstrate how to enhance computational graphs beyond the original input design. This gap risks misleading readers when considering new architecture development. For example, while mixing time is presented as a “good measure,” it tends to increase graph density and does not correlate with improved performance, thus casting doubt on its practical value.

**Other Strengths And Weaknesses:**

### Strengths:

- The presentation is clear and straightforward to follow.
- Analyzing feedforward networks from a graph perspective is an intriguing approach, particularly if it can spur improvements in sparsity or overall performance.
- The theoretical analysis is detailed and appears thorough.

**Questions For Authors:**

- Given that the author's motivations come from analyzing feedforward networks, what is the relation of their analysis to feedforward networks? In particular, given that the nodes are an ordered set, are nodes allowed to have the same order? This would allow to model feedforward networks as seen in [2]

**Relation To Broader Scientific Literature:**

The paper is related to the oversquashing literature from GNNs. Some GNN papers are working on spectral properties/oversmoothing on directed graphs that are missing.

**Theoretical Claims:**

- Their motivation comes from feedforward networks, but their graph models can not accommodate them. For example, given a simple MLP, their line graph model can not simulate the forward pass of MLPs. One would need edge weights. The same holds for other architectures like CNNs, RNNs, or attention blocks.
- There are some unclear parts, e.g., locally connected feedforward graphs with $\kappa=1$ do not have in- and out-degrees equal to $2$ unless every "layer" $n$ (corresponding to order $n$), contains only one node. Please see below for clarifying questions.
- The other proofs appear correct.

---

> ### Author Rebuttal · Authors · 2025-04-01
>
> Dear Reviewer QmmN,
>
> Thank you for your careful review! We hope to provide useful clarifications, and that you may reconsider the relevance of our work:
>
> ### **On feedforward networks**
>
> Your comments focus on our work not easily representing multi-layer feedforward NNs.
>
> For us, the term **“feedforward graph”** is defined more narrowly (cf. Section 3): we study feedforward computation over a _data axis_ (ie. we assume the _input nodes_ have a sequential ordering to them) rather than the _layer axis_! It is hence a _spatial_ constraint rather than one pertaining to multilayer architectures.
>
> Prior work on analysing data-axis feedforward graphs is **rare**: there was no established theory even for single-layer models over such inputs. The entirety of our paper aims to fill this gap, and study how to integrate information across the data axis with a _single_ graph. This corresponds to studying multi-layer GNNs which use the same feedforward graph in every layer. **It was never our intention to study arbitrary multi-layer signal propagation in this paper**.
>
> This was sufficient to fill the page limit and pose important directions for future work. Progressing into the general case (where graphs may differ across layers) is a natural next step, but it induces a combinatorial explosion of design choices. We wanted to understand the basics well before inviting such analyses.
>
> > This gap risks misleading readers when considering new architecture development.
>
> We recognise that, due to the established meaning of “feedforward”, our wording might pose a risk of misleading. We are happy to commit to **changing our title** to be more precise, eg.: “What makes a good _one-step_ feedforward computational graph?”
>
> We also commit to **detailedly discussing** how our setup differs from general multilayer NNs.
>
> ### **On metrics**
>
> Our notion of **mixing time** measures how quickly _salient information_ (ie. input features) can mix within the model. That is, how many feedforward layers (for a given graph) are necessary for a certain level of mixing of salient data to occur.
>
> > ResNets or GPT architectures have ±96 layers to enable the learning of "strong" hidden representations without directly propagating information to output neurons.
>
> This point is _irrelevant_ to our metric, as intermediate neurons in a multi-layer network **do not hold salient information at the beginning**. If every neuron had salient data at the start, then our mixing time would be particularly high; but in the common scenario, our metric’s one-step approach wouldn’t measure mixing in the output neurons only, but at any intermediate point.
>
> On **minimax fidelity**:
>
> > Modern networks routinely prune away large portions of weights or nodes, indicating that focusing on relatively few “important” nodes/edges is advantageous.
>
> The problem is that modern NNs over feedforward data (eg. GPT) are topologically biased towards **earlier nodes** (proved in Barbero et al., NeurIPS’24), and hence not equally well-prepared when important data is not at the start of the input. This is **precisely** what we try to quantify with minimax fidelity: _“which node is in the worst possible position by this choice of graph structure”_?
>
> Fidelity focuses on averaging (rather than pruning) because the mechanisms current models have for pruning, such as the `softmax` function, will provably fail to prune at OOD problem sizes (proved in Veličković et al., 2024).
>
> > In particular, given that the nodes are an ordered set, are nodes allowed to have the same order?
>
> While not explicitly allowed, our framework supports it: simply choose a random order for equal nodes. One component of the FS graph is a block-wise sequence of bipartite communications of this kind.
>
> ### **Further points**
>
> > Related work [1, 2]
>
> We appreciate these relevant papers and will be sure to discuss them in our revision. While [1] generalises oversmoothing to directed graphs, they are not “feedforward”; backwards edges are allowed, making a spectral approach far easier.
>
> > The datasets do not provide meaningful insights into what constitutes a “good” computational graph
>
> We designed our tasks to allow fine-grained control over which (& how many!) nodes are relevant for computing the answer, allowing us to validate our theory’s prescriptions in an unbiased manner.
>
> That said, we agree evaluating real-world setups would make the work more valuable. We have now trained Gemma 2B models – varying the graph used as the attention mask – on the standard Wiki dataset (tensorflow.org/datasets/catalog/wikipedia) containing texts from Wikipedia.
>
> After 3,000 batches of training, our model achieves the following perplexities:
>
> | **Graph**    | **Perplexity** |
> | :------- | :--------: |
> | Full |  $5.3513$   |
> | Line | $5.7267$ |
> | FS | $\mathbf{5.2786}$ |
>
> This trend is consistent throughout training, demonstrating the sparse FS graph is competitive with the full feedforward graph, even when nodes are natural language tokens.

---

> > ### Comment · Reviewer_QmmN · 2025-04-02
> >
> > Dear authors,
> >
> > Thanks for the rebuttal. I indeed misunderstood the motivation—apologies for that. The central question of your work is better phrased as "what makes a good causal attention mask?" in Transformer terms. The term feedforward graph could be confusing, given its association with standard feedforward networks (also often represented as DAGs), which led to my earlier comparison with works like [1]. I agree that, in this light, my earlier critique of the metrics no longer applies.
> >
> > I have updated my review accordingly and raised my score. Some further clarifying questions/points:
> >
> > - Might you consider a “copy-last-token” or "last token copies feature of intermediate token x" tasks (or similar long-range memory benchmark) to highlight whether FS improves upon fully causal masks?
> >
> > - Have you thought about how learned attention weights might partially override low fidelity? You do note that certain theoretical results (e.g. [Barbero et al., 2024]; [Veličković et al., 2024]) question how well attention can prune in the large-length regime, which justifies looking at your purely topological perspective. However, I believe a brief discussion of when attention can adapt enough to mitigate topological constraints would be valuable.
> >
> > - If each layer has a “locally optimal” feedforward graph, where optimal refers to good mixing time and fidelity, does the whole Transformer have a "globally optimal" feedforward graph?
> >
> > - I would like to point to directed expanders discussed in *On directed analogues of expander and hyperfinite
> > graph sequences* by (Csóka and Grabowski, 2021).
> >
> > Best wishes.

---

> > > ### Author Response · Authors · 2025-04-07
> > >
> > > Dear Reviewer QmmN,
> > >
> > > Thank you so much for your positive response and reevaluating your assessment of our work! We are delighted that we’ve been able to reach a mutual understanding about our work’s purpose.
> > >
> > > Your follow-up is full of useful suggestions which are greatly appreciated! Here are our answers:
> > >
> > > > The central question of your work is better phrased as "what makes a good causal attention mask?" in Transformer terms.
> > >
> > > Causal masks are indeed a very relevant deployment direction for our work. We wanted to avoid mentioning _“causal masks”_ in the title as we didn’t want to make any connotations with causality; furthermore, several other important settings – such as temporal graph representation learning – rely on similar spatial constraints but do not require the use of Transformers.
> > >
> > > We will explore the updated wording in the title with careful consideration of all of these points, and in either case the discussion with you was highly informative, and general feedforward graphs are worth discussing in our paper – we will be sure to reference the key takeaways from our exchange prominently in the revision.
> > >
> > > > Might you consider a “copy-last-token” or "last token copies feature of intermediate token x" tasks (or similar long-range memory benchmark) to highlight whether FS improves upon fully causal masks?
> > >
> > > In some sense, we were already doing this (the maximum task requires isolating the maximal token, and then copying one of its features).
> > >
> > > Your suggestion, however, inspired us to test more granularly how such tasks relate to fidelity. For our revision, we have now conducted an experiment where we evaluate on the max task (same as before) but _we **stratify** the reported accuracies as a function of where the max token is._ This allows us to visualise a “performance profile” for the various graph choices – and indeed, the full graph has a sharp collapse towards the last few tokens, while the FS graph remains a consistent predictive power regardless of where the max token is located.
> > >
> > > We will gladly display this in the updated paper, since it shows a clear link between what smaller minimax fidelity predicts (that certain tokens’ representation within the final token’s embedding will eventually “fall off a cliff”) and empirical performance on a task.
> > >
> > > > Have you thought about how learned attention weights might partially override low fidelity? You do note that certain theoretical results (e.g. [Barbero et al., 2024]; [Veličković et al., 2024]) question how well attention can prune in the large-length regime, which justifies looking at your purely topological perspective. However, I believe a brief discussion of when attention can adapt enough to mitigate topological constraints would be valuable.
> > >
> > > We have thought about this a fair bit, and will add some of our key takeaways in the paper.
> > >
> > > Certainly, attention would be capable of overcoming some of the topological restrictions when the length distribution shift is not large enough to trigger the Theorems in the prior works.
> > >
> > > However, we believe that there is another tradeoff that must be contended with: due to the renormalising effect of the softmax function, any attention invested by the modules to “sharpen” less represented paths is attention that is not spent for mixing the relevant information between tokens for answering the task. This tradeoff emphasises the challenges of simultaneously solving the problem given to a Transformer and protecting dataflow along its most vulnerable paths, while also pointing at the fact that alternatives (such as a “pure gate” / sigmoidal attention) might allow for more granularity in this tradeoff.
> > >
> > > > If each layer has a “locally optimal” feedforward graph, where optimal refers to good mixing time and fidelity, does the whole Transformer have a "globally optimal" feedforward graph?
> > >
> > > This is a very interesting question to ponder, and we will highlight it in the future work section.
> > >
> > > This does sound plausible to us, although the definition of “locally optimal” may likely induce some interesting combinations of constraints across layers, which might not be trivial to resolve.
> > >
> > > But we are absolutely certain that research in this space should progress in these kinds of directions, as it is highly unlikely that repeating one layer only will be achieve global optimality in the regime without backwards edges.
> > >
> > > > I would like to point to directed expanders discussed in On directed analogues of expander and hyperfinite graph sequences by (Csóka and Grabowski, 2021).
> > >
> > > Thank you; this is an excellent reference that we will be certain to discuss in our revision!
> > >
> > > Best,
> > > Authors

---

### Official Review · Reviewer_y3Tf · 2025-03-05

**Overall Recommendation:** 3

**Summary:**

The paper studies the impact of feedforward graph structures on information flow, introducing two metrics — mixing time and minimax fidelity— to assess speed and accuracy respectively.

The study reveals a trade-off between fast information propagation and high-fidelity signal propagation among various graph types and proposes the FunSearch (FS) graph generator to design graphs that balance these metrics effectively.

Empirical evaluations highlight that neural networks using the FS graph architecture outperform or match those with traditional designs in both in-distribution and out-of-distribution tasks.

**Claims And Evidence:**

The paper's claims regarding mixing time and minimax fidelity are well-supported by rigorous theoretical analyses in Lemma 4.1, Proposition 5.1, and evidence obtained from evaluations on canonical graph structures in Figure 3.

The introduction of the FS graph generator is backed by theoretical guarantees in Theorem 6.1 and evidence from empirical tests in Figure 3, substantiating the claims of achieving a balance between fast mixing and high fidelity.

However, idealized assumptions limit the generality of the claims (please see 'Theoretical Claims'), while broader applicability remains uncertain without more evidence on real-world datasets (please refer to 'Experimental Designs Or Analyses').

**Essential References Not Discussed:**

Including discussions of random walks on directed acyclic graphs, or connections to causal inference frameworks, could further highlight the novelty of extending these analyses to feedforward graphs.

Discussing recent work on sparse transformer mechanisms, e.g. [1], would enhance this paper by highlighting parallels in balancing computational efficiency and signal preservation, akin to mixing time and minimax fidelity metrics.

A related work [2] presents a novel method that improves long-range connectivity in directed graphs using stochastic rewiring with graph attention.

[1] Big Bird: Transformers for Longer Sequences, In NeurIPS 2020,

[2] Graph Attention with Random Rewiring, arXiv 2407.05649v1.

**Experimental Designs Or Analyses:**

The experiments use synthetic tasks that lack real-world complexity, with graph attention network conditions limited by identity vertex features and synthetic graph structures.

Testing on larger, real datasets with authentic vertex features and real-world graphs would better validate the findings.

Additionally, providing more detailed information on the experimental settings—such as hyperparameter tuning, hyperparameter details of baselines, and statistical significance tests—would enhance the robustness of the findings.

**Methods And Evaluation Criteria:**

The introduction of mixing time and minimax fidelity as complementary metrics measures information propagation efficiency and input detail preservation in graphs.

Theoretical analyses, featuring rigorous derivations and asymptotic studies for various graph constructions, provide a foundation and clarify graph design trade-offs.

The three synthetic tasks — maximum retrieval, second maximum retrieval, and parity computation — test information propagation efficiency and sharpness, effectively stressing different graph aspects and aligning well with the study's objectives.

**Other Comments Or Suggestions:**

The submission would benefit from a discussion on limitations and future research directions.

For example, integrating the proposed metrics with modern design paradigms, such as residual connections and transformer variants, could lead to advanced models.

These models would dynamically adapt their computational graphs based on the specific task.

**Other Strengths And Weaknesses:**

Strengths:

[+] The paper is clearly written and well-organized.

[+] Graph rewiring in the context of feedforward and directed (acyclic) computation graphs is largely unexplored.

Weaknesses:

[-] Experiments are weak; rigorous experiments and ablation studies, e.g., add/remove self-edges, experimenting with various in-degrees, would strengthen the submission.

**Questions For Authors:**

1. Was there a consideration for testing the results on real-world datasets with real-world vertex features? This could demonstrate the practical applicability of the metrics beyond synthetic tasks, increasing confidence in their real-world relevance and utility.
2. Were there considerations for how Theorem 6.1 might adapt to less ideal conditions without strict assumptions like integer-valued log n? This would increase the theorem's applicability to real-world scenarios, enhancing the paper's theoretical impact and generality.
3. Was there a consideration for conducting comprehensive ablation studies, such as adding/removing self-edges? Such studies can improve understanding of experimental design choices, indicating areas for refinement and strengthening future research outcomes.

**Relation To Broader Scientific Literature:**

This submission extends the extensive literature on graph rewiring to feedforward (directed acyclic) graphs — a setting where traditional spectral methods fall short due to the lower-triangular structure.

The introduction of mixing time and minimax fidelity metrics adapts classical Markov chain and expander graph theories to feedforward architectures.

The FS graph generator, inspired by FunSearch, contributes a practical algorithm for designing graphs that balance rapid information mixing with high signal fidelity.

**Theoretical Claims:**

Lemma 4.1 and Proposition 5.1 establish stationary distribution and signal decay in feedforward graphs, with proofs demonstrating valid reasoning under specific assumptions.

However, Theorem 6.1's proof relies on strong assumptions, such as integer-valued log n, perfect divisibility, and fixed minimum fraction of outgoing edges crossing blocks.

Although the overall strategy is plausible and the recurrence analysis leads to a polylogarithmic upper bound, the proof depends on somewhat idealized conditions, raising questions about how robust the result is when these assumptions are relaxed.

---

> ### Author Rebuttal · Authors · 2025-04-01
>
> Dear Reviewer y3Tf,
>
> We are delighted that you have found our foundations to be strong and our results to be interesting. We hope that our responses will strengthen your view of our contributions even further!
>
> > However, Theorem 6.1's proof relies on strong assumptions
>
> Thank you for raising this!
>
> We have now improved the proof of Theorem 6.1 so that all simplifying assumptions have been removed. This turned out to be fairly straightforward, and mostly required the judicious use of ceiling and floor functions; eg. the correct value of the number of levels, $D$, is now:
>
> $$D = \left \lfloor \frac{\log n}{\log \lceil \log n \rceil} \right \rfloor,$$
>
> but the key arguments of our random walk analysis are unchanged.
>
> The proof is unwieldy to paste here but we will of course update it in our revision, and we are happy to discuss further aspects if you would like to know more!
>
> > The experiments use synthetic tasks that lack real-world complexity
>
> We designed our tasks to allow fine-grained control over which (and how many!) nodes are relevant for computing the answer; this allows us to validate our theory’s prescriptions in the most unbiased manner.
>
> That said, we agree that evaluating on real world data would make the work more valuable. We have now trained Gemma 2B models – varying the graph used as the attention mask – on the standard Wiki dataset (https://www.tensorflow.org/datasets/catalog/wikipedia) containing natural language texts scraped from Wikipedia.
>
> After 3,000 batches of training, our model achieves the following perplexities:
>
> | **Graph**    | **Perplexity** |
> | :------- | :--------: |
> | Fully connected |  $5.3513$   |
> | Line | $5.7267$ |
> | FS | $\mathbf{5.2786}$ |
>
> This trend is consistent throughout training. This demonstrates that the sparse FS graph can remain competitive with the full feedforward graph, even when nodes are natural language tokens.
>
> > Providing more detailed information on the experimental settings
>
> Thank you; we will include more details. We used default hyperparameters in most places, and tuned the order-of-magnitude of the learning rate and the weight decay coefficient using the fully-connected graph experiments only (then reused the tuned parameters everywhere else).
>
> > The lack of access to the code as part of the supplementary material limits the ability to verify the experimental results.
>
> We appreciate your remark, and thank you for acknowledging that the paper’s findings should be easier to reproduce since tasks are synthetic. We understand the positive impact of open-sourcing, and would aim to release our code for the synthetic tasks upon acceptance.
>
> > Including discussions of random walks on directed acyclic graphs, or connections to causal inference frameworks, could further highlight the novelty of extending these analyses to feedforward graphs.
>
> We agree that these points are useful, and will add further discussions! To give a concrete example, since our original submission, we managed to make our statement in Section 4.4. (mixing time is ‘small’ iff there are ‘lots’ of paths from ‘most’ vertices to $\tau$) precise:
>
> **Proposition:** _Suppose that the outdegree of every vertex other than $\tau$ is at least $2$. Let $t$ be the average mixing time. Then for some $s \leq t$, the average number of paths from vertex $i$ to $\tau$ with length $s$ is at least $(3/4t)2^s$, where the average is taken over all vertices $i$ between $0$ and $n-1$._
>
> We will include this in the next revision – it provides a relevant connection between our metrics and paths on a DAG and its proof is only a few paragraphs long.
>
> > Related works ([1, 2])
>
> Thank you for bringing these to our attention. They are highly relevant to discuss and we will incorporate them. The key difference is that both Big Bird and GRASS optimise for _bidirectional_ graphs, where back edges are allowed.
>
> > Rigorous experiments and ablation studies, e.g., add/remove self-edges, experimenting with various in-degrees
>
> This is an excellent suggestion! We have now performed both of these ablations, and found that removing self-edges led to significant performance regressions on all tasks – especially on the Parity task where multiple elements need to interact meaningfully for the final answer computation. We also evaluated our graph generators at different levels of in-degree OOM: $O(1)$, $O(\log n)$ and $O(\sqrt{n})$. We find that, as the orders are increased, the models get more performant in-distribution, without a significant effect out-of-distribution. We will incorporate both of these analyses in our revision!
>
> > The submission would benefit from a discussion on limitations and future research directions.
>
> We’ll explicitly include the directions you discussed (integration with Transformer variants, dynamically adapting the computation graph) and we’ll also suggest additional ones (connecting the mixing and fidelity metrics and finding further connections between them and spectrally-motivated analyses!).

---

### Official Review · Reviewer_GmC2 · 2025-03-13

**Overall Recommendation:** 4

**Summary:**

the paper proposed two metrics to measure the quality of computational graph, experiments demonstrate the correlation between those metrics and actual performance.

## update after rebuttal
the author's response addresses my question. the paper looks good to me, I will keep my rating

**Claims And Evidence:**

YES

**Essential References Not Discussed:**

NO

**Experimental Designs Or Analyses:**

YES

**Methods And Evaluation Criteria:**

YES

**Other Comments Or Suggestions:**

NO

**Other Strengths And Weaknesses:**

strength:
1. the paper is well written and easy to follow
2. the experimental part is good, can support the claim

The paper address the challenge of what constitute a good feedforward graph sctructure, especially in terms of information propagation quality. two metrics (mixing time and minimax fidelity) provides us a principle for graph design that could inspire further research in this area, crucial for understanding how effectively information flows through the graph. the experiment parts show that the new graph generator outperforms existing models, showcasing the potential for improved model sparsity and out-of-distribution generalization .

**Questions For Authors:**

1. The paper discusses the asymptotic behaviour of various graphs. why certain structures perform better from an ML perspective.
2. The paper presents empirical results on tasks like maximum retrieval and parity. However, the tasks seem relatively simple. Could the authors expand their evaluation to more complex tasks, such as those involving real-world datasets (e.g., natural language processing or graph-based tasks)?

**Relation To Broader Scientific Literature:**

Related to model optimization, general model training

**Theoretical Claims:**

YES

---

> ### Author Rebuttal · Authors · 2025-04-01
>
> Dear Reviewer GmC2,
>
> We are very thankful for your kind review and recognising the strengths of our work!
>
> To address your questions:
>
> > The paper discusses the asymptotic behaviour of various graphs. why certain structures perform better from an ML perspective.
>
> This is an excellent question. We believe that we can relate at least some of our metrics to established approaches in graph machine learning that study which graphs will have better information propagation from a machine learning perspective.
>
> To name a few examples:
> * **High mixing time** implies that it takes a long time for information to travel to the final vertex (in the worst case – when there is more than one sink – mixing time is infinite). If there are insufficiently many layers in a model to cover these long paths, this directly relates to the well-known _under-reaching_ phenomenon in graph neural networks (Barceló et al., ICLR’20).
> * **Low fidelity** for a particular node implies that this node’s information is not reaching the final node in a way that is not “drowned out” by others, especially as more model layers are added – this effect is well-studied in the GNN literature as the _over-smoothing_ effect, and was recently also studied in the domain of Transformers as well (Barbero et al., NeurIPS’24).
> * High fidelity in certain nodes, but **low _minimax_ fidelity**, implies that, while some nodes are able to sharply send their information into the final node, this is done at the expense of other nodes, which fail to arrive at the sink quickly enough to avoid being overwhelmed. This kind of biased communication is related to the _over-squashing_ effect, which Barbero et al. (NeurIPS’24) have also studied from the perspective of Transformers.
>
> Since our original submission, we have also managed to make our statement in Section 4.4. (that mixing time is ‘small’ if and only if there are ‘lots’ of paths from ‘most’ vertices to $\tau$) precise, as follows:
>
> **Proposition:** _Suppose that the outdegree of every vertex other than $\tau$ is at least $2$. Let $t$ be the average mixing time. Then for some $s \leq t$, the average number of paths from vertex $i$ to $\tau$ with length $s$ is at least $(3/4t)2^s$, where the average is taken over all vertices $i$ between $0$ and $n-1$._
>
> The proof is only a couple of paragraphs long, and it uses the intuition given in Section 4.1. More precisely, $(\mathbf{W}^t)_{\tau i}$ represents the probability of a random walk starting at $i$ and reaching $\tau$ by time $t$. This is equal to the sum, over all $s \leq t$, of the probability that the random walk first reaches $\tau$ at time $s$. For any such $s$, this probability is equal to the sum, over all paths from $i$ to $\tau$ with length $s$ avoiding the loop based at $\tau$, of the probability of taking that path. This latter probability is at most $1/ 2^{s}$ because at each step along the path there were at least two options that could have been taken. So, for some $s \leq t$, the average number of paths from vertex $i$ to $\tau$ with length $s$ is at least $(3/4t)2^s$.
>
> We intend to include this new result in the appendices.
>
> > The paper presents empirical results on tasks like maximum retrieval and parity. However, the tasks seem relatively simple. Could the authors expand their evaluation to more complex tasks, such as those involving real-world datasets (eg. natural language processing or graph-based tasks)?
>
> Thank you for your comment!
>
> We have designed our tasks to allow us fine-grained control over which (and how many!) nodes are relevant for computing the final answer, as this would allow us to validate our theory’s prescriptions in the most unbiased manner.
>
> That being said, we fully agree that evaluating our proposal on real world datasets would make the work more valuable. To this end, we have now trained Gemma 2B language models – varying the graph used as the attention mask – on the standard Wiki dataset (https://www.tensorflow.org/datasets/catalog/wikipedia) containing natural language texts scraped from Wikipedia.
>
> After 3,000 batches of training, our model achieves the following perplexities:
>
> | **Graph**    | **Perplexity** |
> | :------- | :--------: |
> | Fully connected |  $5.3513$   |
> | Line | $5.7267$ |
> | FS | $\mathbf{5.2786}$ |
>
> This trend is consistent throughout training. This demonstrates that the sparse FS graph can remain competitive with the full feedforward graph, even when nodes are natural language tokens.

---

### Decision · Program_Chairs · 2025-05-01

**Decision:**

Accept (poster)

**Comment:**

This paper investigates the impact of feedforward graph structures on information flow, introducing two key metrics: mixing time (to measure propagation speed) and minimax fidelity (to assess signal accuracy). The study reveals a fundamental trade-off between speed and fidelity in different graph topologies and proposes FunSearch (FS), a graph generator designed to balance these objectives. FS is supported by theoretical guarantees (notably Theorem 6.1) and validated through empirical evaluation.

The reviewers appreciated the clarity of presentation, the depth and rigour of the theoretical analysis, and the well-motivated formulation of mixing time and minimax fidelity as complementary performance measures.

The author rebuttal was helpful and thorough, addressing many technical clarifications raised by reviewers. However, concerns remain regarding the experimental validation, which was viewed as less convincing or limited in scope compared to the strength of the theoretical contributions.

Overall, this is a solid theoretical paper with interesting implications, and the introduction of FS with provable properties is a valuable contribution. While the experimental side could be strengthened, the core insights justify inclusion in the program.